# Carnosic Acid Attenuates an Early Increase in ROS Levels during Adipocyte Differentiation by Suppressing Translation of Nox4 and Inducing Translation of Antioxidant Enzymes

**DOI:** 10.3390/ijms22116096

**Published:** 2021-06-05

**Authors:** Dae-Kun Lee, Hae-Dong Jang

**Affiliations:** Department of Food and Nutrition, Hannam University, Daejeon 34504, Korea; ldk7195@naver.com

**Keywords:** carnosic acid, ROS generation, ROS neutralization, Nox4 enzyme translation, NF-κB translocation, IkBα phosphorylation, HO-1 translation, Nrf2 translocation

## Abstract

The objective of this study was to investigate molecular mechanisms underlying the ability of carnosic acid to attenuate an early increase in reactive oxygen species (ROS) levels during MDI-induced adipocyte differentiation. The levels of superoxide anion and ROS were determined using dihydroethidium (DHE) and 2′-7′-dichlorofluorescin diacetate (DCFH-DA), respectively. Both superoxide anion and ROS levels peaked on the second day of differentiation. They were suppressed by carnosic acid. Carnosic acid attenuates the translation of NADPH (nicotinamide adenine dinucleotide phosphate) oxidase 4 (Nox4), p47^phox^, and p22^phox^, and the phosphorylation of nuclear factor-kappa B (NF-κB) and NF-κB inhibitor (IkBa). The translocation of NF-κB into the nucleus was also decreased by carnosic acid. In addition, carnosic acid increased the translation of heme oxygenase-1 (HO-1), γ–glutamylcysteine synthetase (γ-GCSc), and glutathione S-transferase (GST) and both the translation and nuclear translocation of nuclear factor erythroid 2-related factor 2 (Nrf2). Taken together, these results indicate that carnosic acid could down-regulate ROS level in an early stage of MPI-induced adipocyte differentiation by attenuating ROS generation through suppression of NF-κB-mediated translation of Nox4 enzyme and increasing ROS neutralization through induction of Nrf2-mediated translation of phase II antioxidant enzymes such as HO-1, γ-GCS, and GST, leading to its anti-adipogenetic effect.

## 1. Introduction

Obesity is a chronic disease directly caused by abnormal increases in both the size and number of adipocytes. In white adipose tissues, several types of cells, including pre-adipocytes, mature adipocytes, fibroblasts, pericytes, macrophages, neutrophils, lymphocytes, endothelial cells, and adipose stem cells, exist [1]. Pre-adipocyte originating from adipose stem cells are known to be differentiated into adipocytes. This process is called adipogenesis [2]. To control obesity, several strategies such as reducing energy/food intake, increasing energy expenditure, inhibiting adipogenesis, inhibiting lipogenesis, and increasing lipolysis and fat oxidation have been proposed [3]. Among them, decreasing adipogenesis might be one of the better strategies for preventing obesity caused by enhanced adipogenesis [4].

For studying adipogenesis, the murine 3T3-L1 cell line is often used as a reliable in vitro model [5,6,7]. Adipogenesis of 3T3-L1 pre-adipocyte into adipocytes involves the following steps: growth arrest, mitotic clonal expansion, early differentiation, and terminal differentiation [8]. Adipogenesis is a process with tight redox regulation [8,9,10]. Different levels of reactive oxygen species (ROS) are generated in the four steps of adipogenesis of 3T3-L1 pre-adipocytes into adipocytes [8,11]. In particular, a transient increase in ROS levels can facilitate adipogenesis by accelerating mitotic clonal expansion [12]. Based on this evidence, the transient increase in ROS levels, which is non-toxic, can play a role as a transmitter in signaling pathways during differentiation of 3T3-L1 pre-adipocyte into adipocyte [13,14]. Such an increase in ROS levels may occur at the initiation and termination stages of adipogenesis [8].

Carnosic acid is a bioactive compound present in *Rosmarinus officinalis* L. [15]. It has several biological functions including antioxidant [16,17], anti-inflammatory [18,19], anti-bacterial [20], anti-cancer [21,22], and anti-osteoclastic activities [23,24]. In addition, it possesses an anti-adipogenesis activity both in vitro and in vivo [25,26,27]. A recent study has reported that carnosic acid can inhibit the adipogenesis of murine 3T3-L1 pre-adipocytes into adipocytes by down-regulating the redox state and increasing the expression of phase II antioxidant enzyme [28]. However, how carnosic acid attenuates ROS level in the early stage of adipogenesis in aspects of ROS generation and neutralization has not been elucidated.

Thus, the aim of this study was to investigate changes in ROS level during methyl-isobuthylxanthine, dexamethasone, and insulin (MDI)-induced differentiation of 3T3-L1 pre-adipocytes into adipocytes and clarify the molecular mechanisms underlying the down-regulation of the early increase in ROS levels by carnosic acid that led to its anti-adipogenetic effect.

## 2. Results

### 2.1. Carnosic Acid Suppresses Early Increase in Superoxide Anion and ROS Levels during Adipocyte Differentiation

During the differentiation of 3T3-L1 pre-adipocytes into adipocytes, ROS is known to be involved in promoting both early differentiation and later maturation of adipocytes [9]. Therefore, levels of superoxide anion and ROS were monitored using their specific fluorescent probes of dihydroethidium (DHE) and 2′-7′-dichlorofluorescin diacetate (DCFH-DA), respectively. As shown in Figure 1A,B, the intensity of DHE fluorescence was increased at day two, and the intensity of DCF fluorescence was increased at days two and seven during the differentiation. It has been reported that the anti-adipogenic effect of carnosic acid is through ROS control [22]. To determine whether superoxide anion and ROS levels in the early stage of adipocyte differentiation might be down-regulated by carnosic acid, superoxide anion and ROS levels at day two of adipocyte differentiation were analyzed. Carnosic acid at 10–20 μM attenuated the early increase in both superoxide anion and ROS levels were significantly (*p* < 0.001) induced by an MDI hormone mixture (Figure 1C,D). Notably, carnosic acid treatment at 20 μM reduced DCF fluorescence to 125.7%, which was enhanced (163.0%) by the MDI hormone cocktail as compared to an undifferentiated treatment. These observations suggested that the early increase in superoxide anion and ROS levels at day two during the differentiation of 3T3-L1 pre-adipocytes to adipocytes could be abrogated by carnosic acid. These results showed that the potent anti-adipogenetic effect of carnosic acid on the differentiation of 3T3-L1 pre-adipocytes into adipocytes might be attributable to the attenuation of ROS level induced by the MDI hormone cocktail during the early stage of adipocyte differentiation.

### 2.2. Carnosic Acid Attenuates the Translation of Nox4, p47 ^phox^, and p22^phox^ and Nuclear Transport of NF-κB by Inhibiting IκBα Phosphorylation

Nox4 is a multi-component protein that consists of Nox4, p47^phox^, and p22^phox^. It is responsible for superoxide anion generation during MDI-induced differentiation of 3T3-L1 pre-adipocytes to adipocytes [10]. Therefore, the inhibitory effect of carnosic acid on the translation of Nox4 components as Nox4, p47^phox^, and p22^phox^ was analyzed. Results of Western blotting analysis revealed that translations of Nox4, p47^phox^, and p22^phox^ were all significantly (*p* < 0.001) induced by the MDI hormone mixture and notably attenuated by carnosic acid at 1 to 20 μM (Figure 2A–D).

To investigate the involvement of NF-κB/IκBα signal pathway in the translation of Nox4 enzyme components, translation and phosphorylation NF-κB and IκBα with nuclear translocation of NF-κB (p65) levels were analyzed by Western blotting after differentiation for two days (Figure 3A). As shown in Figure 3B, p-NF-κB (p65) in whole cells and nuclei were enhanced by the MDI hormone mixture but obviously attenuated by carnosic acid at 1–20 μM. The suppressive effect of carnosic acid on p-NF-κB in the nucleus was dose-dependent (Figure 3B). This attenuation of NF-κB in the nucleus by carnosic acid was also visually observed by fluorescence imaging using a confocal microscope (Figure 3D). To further make clear the underlying mechanism of transport of NF-κB into the nucleus, the translation and phosphorylation of IκBα, an NF-κB inhibitor, were analyzed by Western blotting. Results showed that IκBα translation was not changed by the MDI hormone mixture or carnosic acid treatment (Figure 3A). However, IκBα phosphorylation was significantly (*p* < 0.001) enhanced by the MDI hormone mixture and diminished by carnosic acid treatment at 1–20 μM (Figure 3C). Taken together, these data suggest that carnosic acid could attenuate the translation of Nox4, p47^phox^, and p22^phox^, the phosphorylation of NF-κB and IκBα, and the nuclear translocation of NF-κB in the early stage of MDI-induced adipocyte differentiation. Accordingly, these results imply that carnosic acid could interrupt the formation of the Nox4 enzyme complex by interfering with the translocation of NF-κB into the nucleus through inhibition of the NF-κB/IκBα signal pathway.

### 2.3. Carnosic Acid Inhibits Adipocyte Differentiation by Attenuating Nox4-Mediated ROS Generation via Interruption of NF-κB/IκBα Signal Pathway

To determine whether the NF-κB/IκBα signal pathway was involved in the down-regulation of Nox4 enzyme expression by carnosic acid, Bay11-7082 (BAY), an inhibitor of IκBα phosphorylation, was used. As shown in Figure 4A, both carnosic acid and BAY diminished the translation of Nox4, p47^phox^, and p22^phox^. As expected, both carnosic acid and BAY attenuated phosphorylation levels of NF-κB and IκBα that were notably enhanced by the MDI hormone mixture (Figure 4B). Furthermore, both carnosic acid and BAY obviously suppressed the translation of CCAAT/enhancer-binding protein α (C/EBPα), C/EBPβ, C/EBPγ, and adiponectin as adipogenesis biomarkers, which were induced by the MDI hormone mixture compared to the undifferentiated treatment (Figure 4C). Taken together, these findings indicate that the translation of Nox4 enzyme components, including Nox4, p47^phox^, and p22^phox^, might be closely associated with the NF-κB/IκBα signal pathway and that the down-regulation of ROS level by carnosic acid in the early stage of adipogenesis might be attributable to the attenuation of Nox4 enzyme translation through interruption of the NF-κB/IκBα signal pathway.

### 2.4. Carnosic Acid induces Nuclear Factor Erythroid 2-Related Factor 2 (Nrf2)-Mediated Translation of Phase II Antioxidant Enzymes

To investigate how carnosic acid down-regulated ROS level in term of ROS neutralization in early stage of MDI-induced adipocyte differentiation, Nrf2-mediated translation levels of phase II antioxidant enzyme as heme oxygenase-1 (HO-1), γ–glutamylcysteine synthetase (γ-GCSc), and glutathione S-transferase (GST) were analyzed by Western blotting. As shown in Figure 5A,B, HO-1 was significantly (*p* < 0.001) reduced by MDI hormone mixture compared to that of an undifferentiated treatment. However, it was enhanced by carnosic acid at 1–20 μM dose-dependently in comparison with the control. On the other hand, protein levels of γ-GCSc and GST is known to be required for glutathione synthesis were induced by the MDI hormone mixture and slightly increased by carnosic acid in comparison with the control (Figure 5C,D). The translation of Nrf2, as a key transcription factor of phase II antioxidant enzyme, was also examined by Western blotting. Results showed that Nrf2 translation was significantly enhanced by MDI hormone mixture and dose-dependently induced by carnosic acid at 1–20 μM in comparison with the control (Figure 6A,B). The translocation of Nrf2 into the nucleus was enhanced by carnosic acid was also confirmed in fluorescence imaging using a confocal microscope (Figure 6C). Collectively, these findings showed that Nrf2-mediated translation of phase II antioxidant enzymes such as HO-1, γ-GCSc, and GST might be induced by carnosic acid, thus contributing to the down-regulation of ROS level through neutralization in the early stage of MDI-induced adipocyte differentiation.

## 3. Discussion

In this study, changes in ROS levels during MDI-induced adipocyte differentiation were monitored using fluorescent probes such as DHE and DCF-DA specific for detecting superoxide anion and ROS, respectively. An early increase in both superoxide anion and ROS levels was observed on the second day of adipocyte differentiation. In addition, a second increase in ROS levels was found on the seventh day of adipocyte differentiation. This is the first study that monitors ROS changes during MDI-induced adipocyte differentiation. The results were in good agreement with previous reports showing a transient ROS increase was induced by insulin in the stage of mitotic clonal expansion or the early stage of the adipocyte differentiation process [8,13].

Based on the above results and the potent anti-adipogenetic effect of carnosic acid, it was hypothesized that carnosic acid could down-regulate the early increase in ROS levels during MDI-induced adipocyte differentiation, thus attenuating the adipogenesis process. As expected, carnosic acid potently abrogated both superoxide anion and ROS levels on the second day of differentiation, although levels of both superoxide anion and ROS were markedly augmented by the MDI hormone mixture. This is the first study reporting that the anti-adipogenetic effect of carnosic acid is due to down-regulation of the early increase in ROS level during MDI-induced adipocyte differentiation. An anti-adipogenetic effect with down-regulation of ROS has also found for caffeic acid phenethyl ester [29], α–lipoic acid [30], (–)-epigallocatechin-3-gallate [31], and natural extracts of *Granteloupia lanceolata* (Okamura) Kawaguchi [32], buckwheat sprouts [33], and unripe kiwi fruit (*Actinidia deliciosa*) [34], although their working mechanisms seem to be different from that of carnosic acid. It is noteworthy that the potent inhibitory effect of carnosic acid on MDI-induced adipogenesis of 3T3-L1 pre-adipocyte cells primarily depends on the attenuation of the early increase in ROS levels. Accumulating evidence has indicated that the redox state during MDI-induced differentiation of pre-adipocytes into adipocytes is regulated by ROS generation through NADPH oxidase 4 (Nox4) and ROS neutralization through antioxidant systems [8]. Therefore, two aspects of ROS generation and neutralization in down-regulation of ROS level by carnosic acid were taken into consideration in this study. Carnosic acid might attenuate ROS generation and/or induce antioxidant systems to down-regulate ROS levels during the early stage of adipocyte differentiation. During adipogenesis, ROS can be primarily generated as hydrogen peroxide because superoxide anion is generated by Nox4, which is highly expressed and immediately converted into hydrogen peroxide by endogenous superoxide dismutase [35]. It has also been suggested that hydrogen peroxide is likely to play a vital role as a signal mediator for cellular functions, including differentiation, because it can easily permeate cells and stay for a longer time than superoxide anion [10,36].

To investigate how carnosic acid down-regulated ROS levels through suppression of ROS generation in the early stage of adipocyte differentiation, suppressive effects of carnosic acid on the translation of Nox4 enzyme components as Nox4, p47^phox^, and p22^phox^ were investigated. According to the results of Western blot analysis, carnosic acid obviously attenuated the translation of Nox4, p47^phox^, and p22^phox^. This implied that carnosic acid might inhibit the formation of active Nox4 enzyme in the early stage of differentiation of 3T3-L1 pre-adipocytes into adipocytes, leading to the attenuation of the early increase in ROS levels. To further understand the underlying mechanism of the inhibitory effect of carnosic acid on Nox4 enzyme translation, the upper signaling pathway, including transcription factors, needs to be investigated. Previous studies have suggested that the translation of the Nox4 enzyme is closely related to redox-sensitive transcription factors such as NF-κB, which is known to play a vital role as a master key in the expression of various proteins as Nox4 during MDI-induced adipogenesis [13,37]. Therefore, the NF-κB/IκBα signal pathway was examined in the present study. NF-κB exists in the cytosol as a complex with IκBα, an inhibitory protein of NF-κB [38]. For the translation of concerned genes including Nox4, p47^phox^, and p22^phox^, NF-κB has to be translocated into the nucleus after IκBα is phosphorylated to be released from the NF-κB/IκBα complex. The reduction in NF-κB translocation into the nucleus by carnosic acid was confirmed by Western blotting and confocal microscopy. The plausible explanation for this observation was that carnosic acid could inhibit IκBα phosphorylation, leading to the maintenance of NF-κB/IκBα complex and the decrease in NF-κB due to degradation by the proteosome. The inhibition of IκBα phosphorylation by carnosic acid was checked using BAY as an inhibitor of IκBα phosphorylation. As shown in Figure 4, both carnosic acid and Bay treatment markedly suppressed the translation of Nox4, p47^phox^, p22^phox^, adiponectin, and NF-κB as well as the phosphorylation of IκBα and NF-κB, which were enhanced by MDI hormone mixture as compared to the undifferentiated treatment. These results implied that carnosic acid might interrupt IκBα phosphorylation to decrease NF-κB levels in the nucleus, which could cause attenuated translation of Nox4, p47^phox^, and p22^phox^. In this study, how carnosic acid inhibited IκBα phosphorylation was not elucidated. However, recent studies have proposed that catechol-type polyphenols could be oxidized into electrophilic quinones by ROS transiently and temporarily increased in cell [39,40,41]. Therefore, carnosic acid carrying catechol moiety might be oxidized by ROS produced during the early stage of adipocyte differentiation into its quinone and interact with sensible sulfhydryl groups of cysteine residues existing on the surface of IκBα and/or its kinase, leading to their conformational changes, which might be associated with an interruption of IκBα phosphorylation.

With respect to ROS neutralization by carnosic acid in the early stage of adipocyte differentiation, induction of the antioxidant system by carnosic acid was confirmed. HO-1 is known to be one important antioxidant enzyme involved in down-regulating ROS during the early stage of adipogenesis [42,43], and γ-GCS and GST are key enzymes in glutathione synthesis [28], these three antioxidant enzymes were chosen in this study. HO-1 was markedly attenuated in an early stage of MDI-induced adipocyte differentiation while ROS level was increased, consistent with earlier reports showing the import role of HO-1 in ROS neutralization during the early stage of adipogenesis [44,45]. However, HO-1 was notably induced by carnosic acid, which was closely related to the strong suppressive effect of carnosic acid on ROS level. On the other hand, γ-GCSc, and GST were notably induced by the MDI hormone mixture and slightly increased by carnosic acid compared to controls. These results imply that ROS levels in the early stage of adipogenesis might depend on HO-1, and that could be down-regulated through induction of HO-1 translation by carnosic acid. For further understanding of the induction of HO-1, γ-GCSc, and GST translation by carnosic acid, Nrf2/keap1 signaling pathway was investigated by Western blot and confocal microscopy. Both translation and nuclear translocation of Nrf2 were slightly enhanced by the MDI hormone mixture, leading to substantial increases of γ-GCSc, and GST translation, which might be necessary for the expression of some proteins needed in the early stage of adipogenesis [28]. On the other hand, the translation and nuclear translocation of Nrf2 were remarkably induced by carnosic acid, which might lead to the translation of HO-1 rather than γ-GCSc or GST. Consequently, carnosic acid can down-regulate ROS level during the early stage of adipocyte differentiation by neutralizing ROS through Nrf2-mediated induction of phase II antioxidant enzymes such as HO-1, γ-GCSc, and GST.

## 4. Materials and Methods

### 4.1. Chemical and Reagents

Carnosic acid was purchased from Cayman Chemical (Ann Arbor, MI, USA). Fetal bovine serum (FBS) was obtained from Biowest (Riverside, MO, USA). Dulbecco`s modified Eagle`s medium (DMEM), bovine calf serum (BCS), fetal bovine serum (FBS), trypsin-EDTA, phosphate-buffered saline (PBS, pH 7.4), and Hank`s balanced salt solution (HBSS) were purchased from Welgene Inc. (Gyeongsan, Korea). Dimethylsulfoxide (DMSO), antibiotic-antimycotic solution 100X, insulin, 3-isobutyl-1-methyl-xanthine (IBMX), dexamethasone, isopropanol, formaldehyde, DCFH-DA, DHE, protease inhibitor, phenylmethanesulfonyl fluoride (PMSF), Triton X-100, 1,4-dithiothreitol (DTT), skim milk, and DPAI (4′,6-diamidino-2-phenylindole) were purchased from Sigma-Aldrich (St. Louis, MO, USA). Antibodies of Nox4, p47^phox^, p22^phox^, NF-κB (p65), phospho-NF-κB (p65), IκBα, phospho-IκBα, C/EBPα, C/EBPβ, C/EBPγ, HO-1, γ-GCSc, GST, and Nrf2 were purchased from Santa Cruz Biotechnology (Santa Cruz, CA, USA). Anti-adiponectin was obtained from Cell Signaling Technology (Beverly, MA, USA). 3T3-L1 cells were purchased from the American Type Culture Collection (ATCC, Rockville, MD, USA).

### 4.2. Cell Culture and Differentiation

The differentiation of pre-adipocytes into adipocytes was performed by a published procedure [46]. Briefly, 3T3-L1 pre-adipocyte cells were purchased from ATCC and grown to confluence in six-well plates containing DMEM supplemented with 10% BCS and antibiotics (100 U/mL penicillin and 100 μg/mL streptomycin) under 5% CO_2_ humidified atmosphere at 37 °C. After two days of culturing, cells reached confluence, and the medium was replaced with a differentiation medium containing 10% FBS and MDI (0.5 mM 1-methy-3-isobuthyllxanthine, 1 μM dexamethasone, and 5 μg/mL insulin) hormone mixture [47]. Two days later, the differentiation medium was replaced with DMEM medium supplemented with 10% FBS and 5 μg/mL insulin. After two days, the medium was changed to DMEM supplemented with 10% FBS every two days until day 8 of differentiation. Carnosic acid was added two days before cells reached confluence and kept for four days until the second day.

### 4.3. Determination of Intracellular Superoxide Anion and ROS

Fluorescent probes, DCFH-DA (Ex/Em = 485 nm/535 nm) and DHE (Ex/Em = 518 nm/605 nm) that could specifically react with ROS and superoxide anion, respectively, were used to check ROS and superoxide anion levels during the differentiation of 3T3-L1 cells into adipocyte for eight days and to investigate the suppressive effect of carnosic acid on ROS and superoxide anion [48]. Briefly, 3T3-L1 cells were seeded into 96-well plates (1 × 10^5^ cells/well) containing DMEM plus 10% BCS and antibiotics and incubated until cells reached confluence. After the confluence of cells was reached, the medium was replaced with DMEM supplemented with 10% BCS and carnosic acid. Two days later, the medium was then replaced with the differentiation medium. After 1 to 8 days of incubation, the medium was removed, and wells were washed twice successively with PBS and HBSS. A fluorescent probe was then added to the culture plates at a final concentration of 50 μM. After incubating in the dark for 30 min at 37 °C, fluorescence intensities of DHE and DCFH-DA were measured using a fluorometric plate reader (SpectraMax i3, Molecular Devices, San Jose, CA, USA).

### 4.4. Western Blot Analysis

3T3-L1 cells were seeded into 60 mm cell culture dishes (1 × 10^5^ cells/dish) containing DMEM medium plus 10% BCS and antibiotics and incubated until cells reached confluence. After the confluence of cells was reached, the medium was replaced with DMEM containing 10% BCS and carnosic acid. Two days later, the medium was replaced with a differentiation medium. After incubation for another two days, cells were washed twice with PBS. 3T3-L1 cells were then lysed with ice-cold radio-immunoprecipitation assay (RIPA) buffer which was composed of 50 mM Tris–HCl (pH 8.0), 1% NP-40, 0.5% sodium deoxycholate, 150 mM NaCl, 1 mM PMSF, and a protease inhibitor cocktail [49]. The protein from lysed cells was then subjected to sodium dodecylsulfate-polyacrylamide gel electrophoresis (SDS-PAGE) and transferred to nitrocellulose membranes. Membranes were blocked with 5% skimmed milk in Tris-buffered saline containing Tween 20 (TBST) for 1 h at room temperature and then incubated with primary antibodies overnight at 4 °C. Proper horseradish peroxide-conjugated secondary antibodies were then added and incubated with the membranes for 1 h at room temperature. Proteins present in the membranes were visualized with an EZ-Western Lumi pico detection kit (DoGEN, Seoul, Korea) and quantified using a FUSION SOLO S (Vilber Lourmat, Collégien, France).

### 4.5. Nuclear Protein Extraction

Nuclear protein extraction was performed using a nuclear extraction kit (Abcam, Cambridge, UK). The harvested cell suspension was centrifuged at 300× *g* for 5 min. The supernatant was carefully removed and discarded. Afterward, 0.15 mL of a 1X pre-extraction buffer was added to the cell pellet followed by vigorous vortexing for 10 s, incubation on ice for 10 min, and centrifugation at 15,800× *g* for 1 min to remove cytosolic proteins from the nuclear pellet. The supernatant which contained the cytosolic protein was removed and transferred to a new tube (cytosol protein). Then, 0.06 mL of extraction buffer containing DTT and protease inhibitor cocktail (PIC) was added to the nuclear pellet. The extraction mixture was incubated on ice for 15 min. It was vortexed for 5 s every 3 min. The suspension was then centrifuged at 18,000× *g* for 10 min at 4 °C. The supernatant was then transferred into a new microcentrifuge vial as a nuclear protein extract.

### 4.6. Confocal Microscopy

Microscopical observations of cells were performed by confocal imaging as described previously [50]. Briefly, following adipocyte differentiation for 2 days, 3T3-L1 cells were washed twice with PBS and fixed with 3.7% paraformaldehyde in PBS for 15 min at room temperature. After cells were washed with PBS and incubated with PBS containing 0.2% Triton X-100 at room temperature for 10 min, they were blocked with 3% skimmed milk in PBS for 45 min at room temperature. The blocking solution was removed, and the remaining cells were incubated with 1:250 solution of p-NF-κB primary antibody in 3% skimmed milk overnight at 4 °C. Cells were then washed three times with PBS (5 min each wash) and incubated with a 1:2500 solution of AlexaFlour 546 conjugated goat anti-rabbit secondary antibody (Invitrogen, Carlsbad, CA, USA) in a dark condition for 45 min at room temperature. Cells were washed three times with PBS (5 min each wash). A LSM5 live configuration Vario Two VRGB microscope (Zeiss, Jena, Germany) equipped with an oil immersion lens was used for the imaging of cells. Fluorescence imaging was obtained using 405 nm and 535 nm lasers for detecting nuclear staining of DAPI and anti-p-NF-κB (p65), respectively.

### 4.7. Statistical Analysis

All data are presented as mean ± SEM. All statistical analyses were executed using the statistical package SPSS 11 program (Statistical Package for Social Science 11, SPSS Inc., Chicago, IL, USA). The statistical significance for each group was verified with one-way analysis of variance (ANOVA) followed by Duncan’s test at *p* < 0.05 or Student’s *t*-test (^#^: *p* < 0.05; ^##^: *p* < 0.01; ^###^: *p* < 0.001).

## 5. Conclusions

During MDI-induced differentiation of 3T3-L1 pre-adipocytes into adipocytes, ROS levels peaked on the second day and could be potently attenuated by carnosic acid treatment. Carnosic acid could down-regulate ROS levels during the early stage of MPI-induced adipocyte differentiation by attenuating ROS generation through suppression of NF-B mediated translation of Nox4 enzyme and increasing ROS neutralization through induction of Nrf2-mediated translation of phase II antioxidant enzymes such as HO-1, γ-GCSc, and GST, leading to its anti-adipogenetic effect.

## Figures and Tables

**Figure 1 ijms-22-06096-f001:**
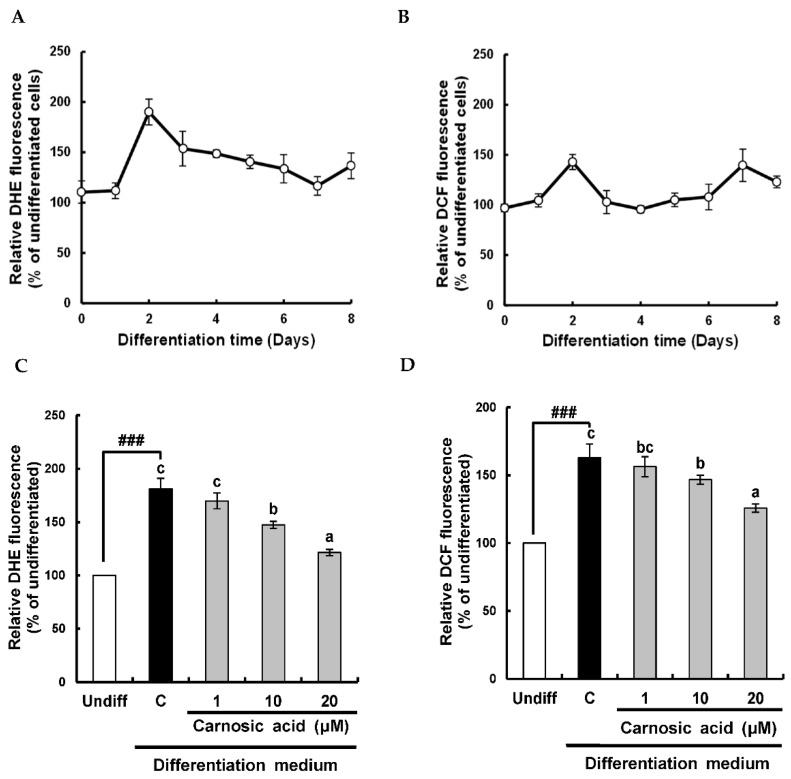
Carnosic acid suppresses superoxide anion and ROS levels in MDI-induced differentiation of 3T3-L1 pre-adipocytes into adipocytes. (**A**) Change of superoxide anion levels during adipogenesis. (**B**) Change of ROS levels during adipogenesis. (**C**) Carnosic acid suppressed superoxide anion levels after differentiation for two days. (**D**) Carnosic acid suppressed ROS levels after differentiation for two days. Data are presented as averages of three independent experiments performed in triplicates and expressed as percentages of the value of the control (means ± standard error mean, *n* = 3). ^###^: *p* < 0.001 vs. Undiff. Different corresponding letters indicate significant differences at *p* < 0.05 using Duncan’s test. (Undiff: undifferentiated, which was not treated with the MDI hormone mixture; C: control, which was treated with the MDI hormone mixture).

**Figure 2 ijms-22-06096-f002:**
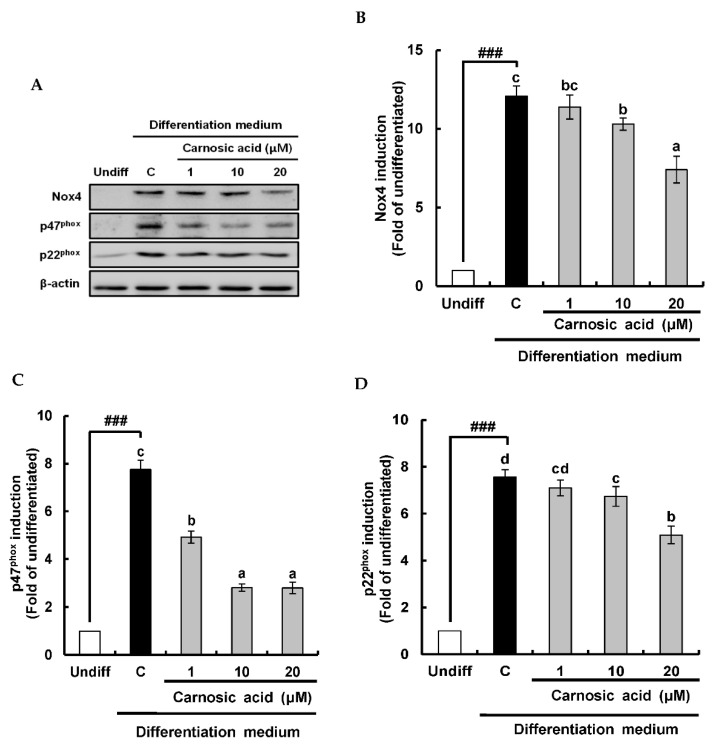
Inhibitory effect of carnosic acid on the translation of Nox4 enzyme components in MDI-induced differentiation of 3T3-L1 pre-adipocytes to adipocytes for two days. (**A**) Translation of Nox4, p47^phox^, and p22^phox^ as determined by Western blotting analysis after differentiation for two days. (**B**) Graphic representation of densitometric analysis of Nox4 blot. (**C**) Graphic representation of densitometric analysis of p47^phox^ blot. (**D**) Graphic representation of densitometric analysis of p22^phox^ blot. Data are presented as averages of three independent experiments performed in triplicates and expressed as percentages of the value of control (means ± standard error mean, *n* = 3). ^###^: *p* < 0.001 vs. Undiff. Different corresponding letters indicate significant differences at *p* < 0.05 using Duncan’s test. (Undiff: undifferentiated, which was not treated with the MDI hormone mixture; C: control, which was treated with the MDI hormone mixture).

**Figure 3 ijms-22-06096-f003:**
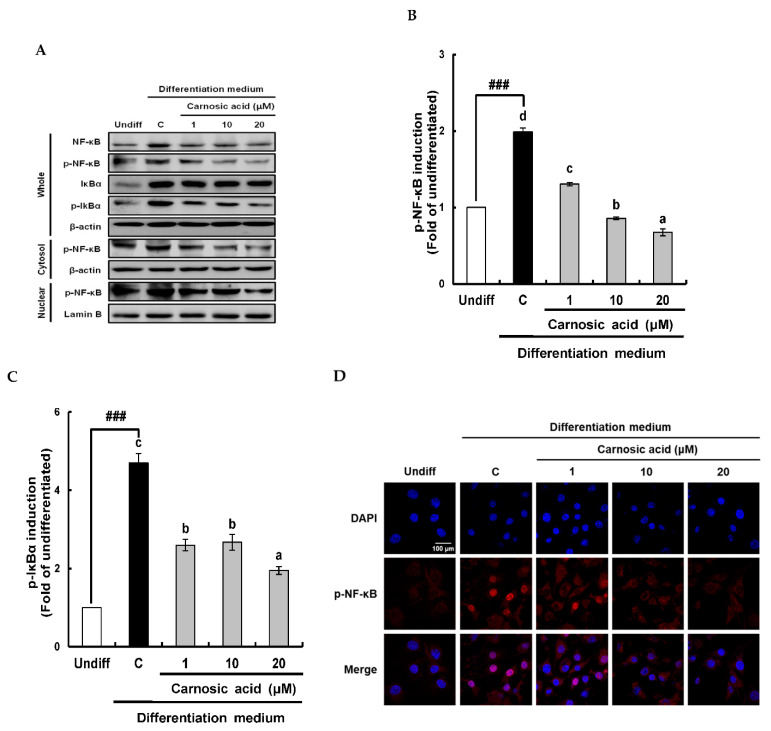
Inhibitory effect of carnosic acid on translation and phosphorylation of NF-κB and IκBα, and translocation of NF-κB into the nucleus during the differentiation of MDI-induced 3T3-L1 pre-adipocytes into adipocytes for two days. (**A**) Translated and phosphorylated NF-κB and IκBα in whole cells by Western blotting analysis after differentiation for two days. (**B**) Graphic representation of densitometric analysis of nuclear p-NF-κB blot. (**C**) Graphic representation of densitometric analysis of p-IκBα blot. (**D**) Immunofluorescence staining of 3T3-L1 pre-adipocyte cells after treatment with 1–20 μM carnosic acid using anti-p-NF-κB (p65) for staining. Data are presented as averages of three independent experiments performed in triplicates and expressed as percentages of the value of the control (means ± standard error mean, *n* = 3). ^###^: *p* < 0.001 vs. Undiff. Different corresponding letters indicate significant differences at *p* < 0.05 using Duncan’s test. (Undiff: undifferentiated, which was not treated with the MDI hormone mixture; C: control, which was treated with MDI hormone mixture; DAPI: 4′,6-diamidino-2-phenylindole).

**Figure 4 ijms-22-06096-f004:**
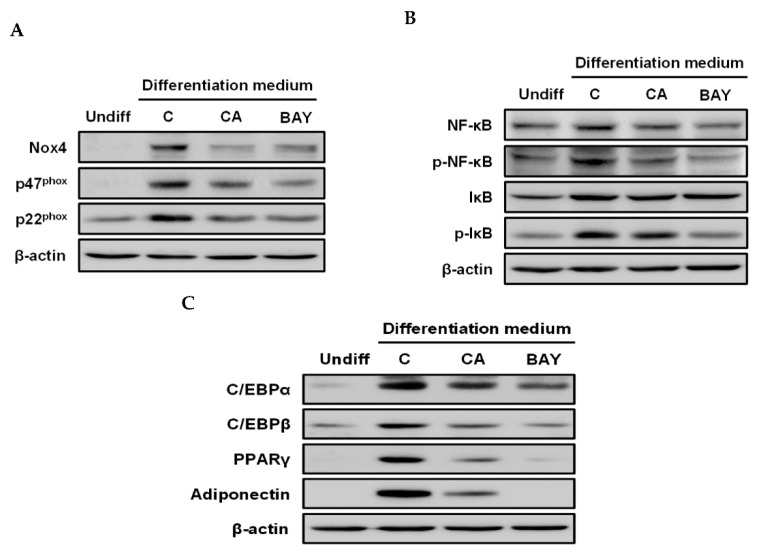
Effects of carnosic acid on the translation of Nox4 enzyme components, phosphorylation of NF-κB and IκBα, and translation of adipogenesis biomarkers in MDI-induced differentiation of 3T3-L1 pre-adipocytes to adipocytes for two days. (**A**) Translation levels of Nox4, p47^phox^, and p22^phox^ by Western blotting analysis after differentiation for two days. (**B**) Translation and phosphorylation levels of NF-κB and IκBα by Western blotting analysis after differentiation for two days. (**C**) Translation levels of C/EBPα, C/EBPβ, C/EBPγ, and adiponectin by Western blotting analysis after differentiation for two days. Data are presented as representative images of three independent experiments performed in triplicates). (Undiff; undifferentiated, which was not treated with MDI hormone mixture; C: control, which was treated with MDI hormone mixture; BAY: Bay11-7082).

**Figure 5 ijms-22-06096-f005:**
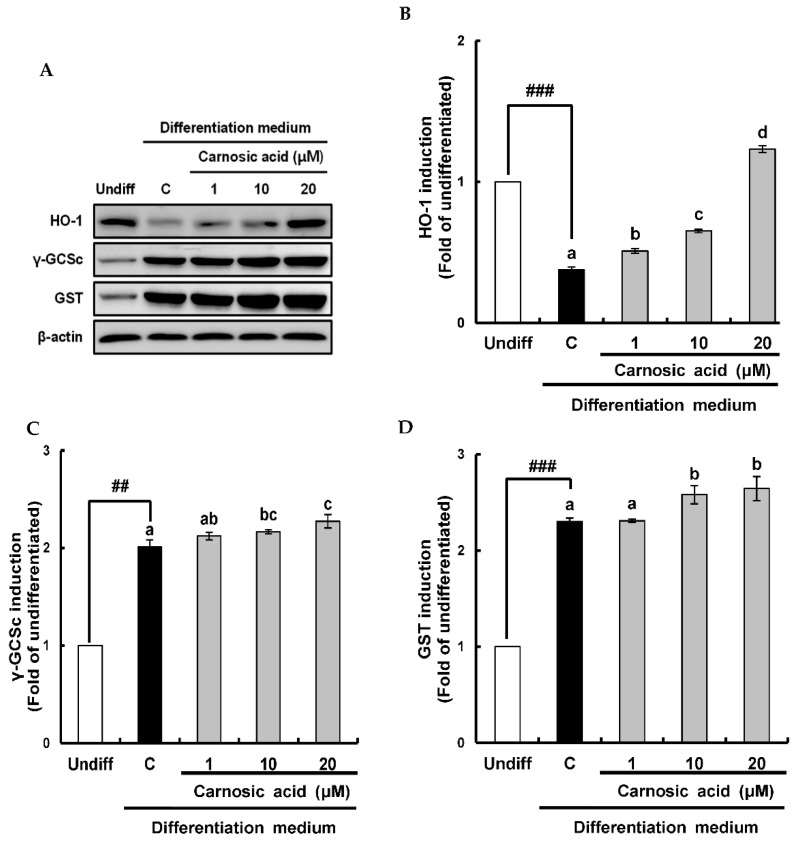
Effects of carnosic acid on the translation of phase II antioxidant enzyme during MDI-induced differentiation of 3T3-L1 pre-adipocytes to adipocytes for two days. (**A**) Translation of phase II antioxidant enzyme by Western blotting analysis after differentiation for two days. (**B**) Graphic representation of densitometric analysis of HO-1 blot. (**C**) Graphic representation of densitometric analysis of phosphorylated γ-GCSc blot. (**D**) Graphic representation of densitometric analysis of GST blot. Data are presented as averages of three independent experiments performed in triplicates and expressed as percentages of the value of the control (means ± standard error mean, *n* = 3). ^##^: *p* < 0.01 and ^###^: *p* < 0.001 vs. Undiff. Different corresponding letters indicate significant differences using Student’s *t*-test. (Undiff: undifferentiated, which was not treated with the MDI hormone mixture; C: control, which was treated with the MDI hormone mixture).

**Figure 6 ijms-22-06096-f006:**
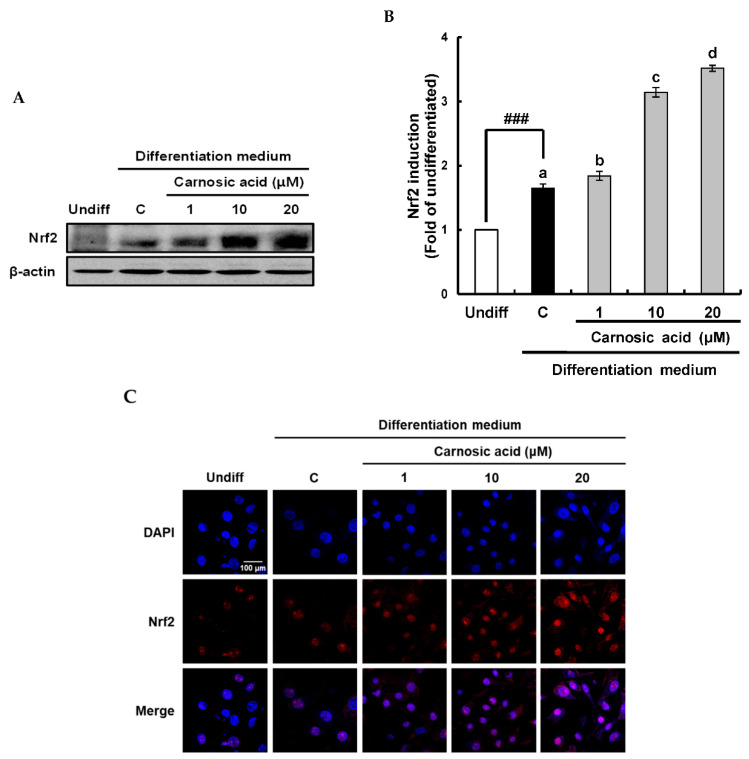
Effects of carnosic acid on translation and nuclear translocation of Nrf2 during MDI-induced differentiation of 3T3-L1 pre-adipocytes to adipocytes for two days. (**A**) Translation of Nrf2 by Western blotting analysis after differentiation for two days. (**B**) Graphic representation of densitometric analysis of Nrf2 blot. (**C**) Immunofluorescence staining of 3T3-L1 pre-adipocyte cells subjected to 1–20 μM carnosic acid using anti-Nrf2. Data are presented as averages of three independent experiments performed in triplicates and expressed as percentages of the value of the control (means ± standard error mean, *n* = 3). ^###^: *p* < 0.001 vs. Undiff. Different corresponding letters indicate significant differences by Student’s *t*-test. (Undiff: undifferentiated, which was not treated with the MDI hormone mixture; C: control, which was treated with the MDI hormone mixture; DAPI: 4′,6-diamidino-2-phenylindole).

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
