# Peer review of "Carnosic Acid Attenuates an Early Increase in ROS Levels during Adipocyte Differentiation by Suppressing Translation of Nox4 and Inducing Translation of Antioxidant Enzymes"

_ijms, 2021, doi:10.3390/ijms22116096_

Round 1

Reviewer 1 Report

Here I report my review comments on the article of Dae-Kun Lee and Hae-Dong Jang.

In this experimental work the Authors give interesting data about the ability of carnosic acid to decrease ROS level in early stage of adipocyte differentiation. In addition, the Authors described the molecular mechanisms hypothesizing that the NF-κB and Nrf2 pathways could be involved in the mechanism behind the ROS downmodulation in 3T3-L1 pre-adipocytes.  The research in literature did not show any similar paper like this one, and, based on the importance of this topic, I consider that the manuscript can be accepted for publication on IJMS. Title is too long. Would it be possible to reduce the length of the title? the text is writing in good English without typos.

Minor issues:

Abstract, line 11: Authors report the abbreviation “MPI”. Is there an error? the abbreviation is MDI? Please correct.

Keywords, line 25: delete the word “keywors 1” in the list

Line 61: correct “ealy” with “early”

Lines 105 and 195: delete the extra round brackets.

Figure 2: I recommend the Authors to insert the symbols indicating significant differences in all bars in the histograms in figure 2.

Line 354, 4.2 Cell culture and differentiation: The Authors cultured the 3T3-L1 pre-adipocytes in differentiation medium containing serum and MDI hormone mixture for 2 days to induce the differentiation. “Two days later, differentiation medium was replaced with medium containing 10% FBS and 5 μg/mL insulin. After two days, the medium was changed with 10% FBS medium every two days until day 8 of differentiation.” How do the Authors know that two days are enough to induce adipocyte differentiation? How do they evaluate any morphological or genetic changes in differentiated cells? Did they use Oil red O staining or did they perform PCR on specific genes? Please specify.

Statistical analysis: Authors reported “All data were presented as mean ± SD” but it would be more correct to use the standard error mean (SEM) instead of standard deviation (SD) if the experiments are independent.

To improve the quality of this manuscript, Authors should replace and/or insert more recent bibliographic references throughout all the text.

 Did the Authors evaluate the effect of Carnosic acid on mesenchymal stem cell derived from adipose tissue? I believe that it could be interesting to debate about it in the Discussion section.

Kind regards

Author Response

In this experimental work the Authors give interesting data about the ability of carnosic acid to decrease ROS level in early stage of adipocyte differentiation. In addition, the Authors described the molecular mechanisms hypothesizing that the NF-κB and Nrf2 pathways could be involved in the mechanism behind the ROS downmodulation in 3T3-L1 pre-adipocytes.  The research in literature did not show any similar paper like this one, and, based on the importance of this topic, I consider that the manuscript can be accepted for publication on IJMS. Title is too long. Would it be possible to reduce the length of the title? the text is writing in good English without typos.

Response: Thank you for your suggestion. Title is shortened like “Carnosic Acid Attenuates an Early Increase of ROS Level During Adipocyte Differentiation by Suppressing Translation of Nox4 and Inducing Translation of Antioxidant Enzymes”.

Minor issues:

Abstract, line 11: Authors report the abbreviation “MPI”. Is there an error? the abbreviation is MDI? Please correct.

Response: “MPI” is replaced with “MPI”.

Keywords, line 25: delete the word “keywors 1” in the list

Response: “keywors 1” is deleted.

Line 61: correct “ealy” with “early”

Response: “ealy” is replaced with “early”.

Lines 105 and 195: delete the extra round brackets.

Response: The extra round brackets are deleted.

Figure 2: I recommend the Authors to insert the symbols indicating significant differences in all bars in the histograms in figure 2.

Response: The symbols indicating significant differences are already inserted in all bars in Figure 2

Line 354, 4.2 Cell culture and differentiation: The Authors cultured the 3T3-L1 pre-adipocytes in differentiation medium containing serum and MDI hormone mixture for 2 days to induce the differentiation. “Two days later, differentiation medium was replaced with medium containing 10% FBS and 5 μg/mL insulin. After two days, the medium was changed with 10% FBS medium every two days until day 8 of differentiation.” How do the Authors know that two days are enough to induce adipocyte differentiation? How do they evaluate any morphological or genetic changes in differentiated cells? Did they use Oil red O staining or did they perform PCR on specific genes? Please specify.

Response: The differentiation of 3T3-L1 pre-adipocyte into adipocyte was performed by procedure as described by reference 41 (He, Y.; Li, Y.; Zhao, T.; Wang, Y.; Sun, C. Ursolic acid inhibits adipogenesis in 3T3-L1 adipocytes through LKB1/AMPK pathway. PLos ONE E 2013, 8, e70135.). According to reference 41, the treatment with MDI hormone mixture for two days is enough for inducing adipocyte differentiation.

Statistical analysis: Authors reported “All data were presented as mean ± SD” but it would be more correct to use the standard error mean (SEM) instead of standard deviation (SD) if the experiments are independent.

Response: Thank you for your comments. “SD” is changed to “SEM”.

To improve the quality of this manuscript, Authors should replace and/or insert more recent bibliographic references throughout all the text.

Response: According to your comments, old references are replaced with recent bibliographic references and more references are inserted.

 Did the Authors evaluate the effect of Carnosic acid on mesenchymal stem cell derived from adipose tissue? I believe that it could be interesting to debate about it in the Discussion sectiKind regards

Response: In this study, the effect of carnosic acid on mesenchymal stem cell derived from adipose tissue is not evaluated. In the next study using animal model, we are going to do it.

Reviewer 2 Report

Authors submitted a research article titled Carnosic acid Attenuates the Early Rise in ROS Level during Adipocyte Differentiation through Suppression of NF-κB-Mediated Translation of Nox4 Enzyme and Induction of Nrf2-Mediated Translation of Antioxidant enzyme. Very interesting finding and provides useful information about Carnosic acid in free radical reduction and inhibition of adipogenesis in 3T-3L1 cells. The experimental protocol and the experimentation were acceptable. Data generation and its interpretation are good. 

The background details in the introduction are not sufficient. They must improve the introduction section by adding necessary information in sequential order such as adipocyte differentiation, obesity and its complication, mechanisms between ROS and obesity, and compound.

Line numbers 69-69 please rewrite clearly. 

Line number 74-77 authors stated Carnosic treatment at 50uM reduced DCL level; I could not see the 50uM treated data in any figures. Please check it

Line number 78-80, the potent adipogenic effect of carnosic acid on 3T3-L1 pre-adipocyte differentiation into adipocyte may be attributable to the attenuation of ROS level, which is induced by MDI hormone cocktail in an early stage of adipocyte differentiation. Does carnosic acid induce adipogenesis? 

The authors determined ROS level by MDI in cells for seven days. How about a compound treatment schedule?  

Carnosic acid inhibits the adipocyte differentiation effect by attenuating Nox4-mediated ROS 162 generation via interruption of NF-κB/IκBα signal pathway. If possible, provide lipid accumulation data in control and compound treated cells.

.

Author Response

Authors submitted a research article titled Carnosic acid Attenuates the Early Rise in ROS Level during Adipocyte Differentiation through Suppression of NF-κB-Mediated Translation of Nox4 Enzyme and Induction of Nrf2-Mediated Translation of Antioxidant enzyme. Very interesting finding and provides useful information about Carnosic acid in free radical reduction and inhibition of adipogenesis in 3T-3L1 cells. The experimental protocol and the experimentation were acceptable. Data generation and its interpretation are good. 

Response: Thank you for your comments.

The background details in the introduction are not sufficient. They must improve the introduction section by adding necessary information in sequential order such as adipocyte differentiation, obesity and its complication, mechanisms between ROS and obesity, and compound.

Response: In our study, we tried to investigate how carnosic acid can attenuate the early increase of ROS level during adipocyte differentiation. Therefore, according to your suggestion, introduction section is rearranged in sequential order as obesity, adipocyte differentiation and its complication, ROS in adipocyte differentiation, and carnosic acid.

Line numbers 69-69 please rewrite clearly.              

Response: “To investigate whether the early rise in superoxide anion and ROS level can be involved in inhibitory effect of carnosic acid on MDI-induced adipogenesis of 3T3-L1 pre-adipocyte cells, the effect of carnosic acid treatment on the early rise of superoxide anion and ROS level was analyzed.” is replaced with “It has been reported that the anti-adipogenic effect of carnosic acid is through ROS control [22]. To determine whether superoxide anion and ROS levels in the early stage of adipocyte differentiation might be down-regulated by carnosic acid, superoxide anion and ROS levels at day two of adipocyte differentiation were analyzed.“.

Line number 74-77 authors stated Carnosic treatment at 50uM reduced DCL level; I could not see the 50uM treated data in any figures. Please check it

Response: You are right. 50 uM is changed to 20 uM.

Line number 78-80, the potent adipogenic effect of carnosic acid on 3T3-L1 pre-adipocyte differentiation into adipocyte may be attributable to the attenuation of ROS level, which is induced by MDI hormone cocktail in an early stage of adipocyte differentiation. Does carnosic acid induce adipogenesis? 

Response: “the potent adipogenic effect of carnosic acid’ is replaced with ‘the anti-adipogenic effect of carnosic acid”

The authors determined ROS level by MDI in cells for seven days. How about a compound treatment schedule?  

Response: ROS was determined for eight days and two increases of ROS level were found on the second and seventh day. In this study, we are concerned the ROS increase on the second day. The cells were treated with MDI hormone mixture after confluence. Carnosic acid was added at two days before cells reached confluence and kept for 4 days until the second day. 

Carnosic acid inhibits the adipocyte differentiation effect by attenuating Nox4-mediated ROS 162 generation via interruption of NF-κB/IκBα signal pathway. If possible, provide lipid accumulation data in control and compound treated cells.

Response: Your suggestion looks like being good. But the effect of carnosic acid on lipid accumulation was already reported in reference 19 (Park, M.Y.; Sung, M.K. Carnosic acid inhibits lipid accumulation in 3T3-L1 adipocytes and through attenuation of fatty acid desaturation. J. Cancer Prev. 2015, 20, 41–49.).

Round 2

Reviewer 2 Report

The authors have submitted a revised research article. I have gone through the whole manuscript thoroughly. The authors have made significant changes in the introduction section and rectified the errors mentioned by the reviewer. This manuscript shows substantial improvement than the previously submitted. This paper can be suitable for acceptance in IJMS in the present format